# Uptake Quantification of Antigen Carried by Nanoparticles and Its Impact on Carrier Adjuvanticity Evaluation

**DOI:** 10.3390/vaccines12010028

**Published:** 2023-12-26

**Authors:** Yupu Zhu, Minxuan Cui, Yutao Liu, Zhengjun Ma, Jiayue Xi, Yi Tian, Jinwei Hu, Chaojun Song, Li Fan, Quan Li

**Affiliations:** 1Department of Pharmaceutical Chemistry and Analysis, School of Pharmacy, Airforce Medical University, 169th Changle West Road, Xi’an 710032, China; zhuyupupapa@fmmu.edu.cn (Y.Z.); cuiminxuan2521@fmmu.edu.cn (M.C.); liuyutao@fmmu.edu.cn (Y.L.); mazhengjun@fmmu.edu.cn (Z.M.); 0518xjy@fmmu.edu.cn (J.X.); hujinwei@fmmu.edu.cn (J.H.); 2Department of Oncology, Airforce Medical Center of PLA, 30th Fu Cheng Road, Beijing 100142, China; tiannyii@163.com; 3School of Life Science, Northwestern Polytechnical University, 127th Youyi West Road, Xi’an 710072, China; cj6005@nwpu.edu.cn; 4Department of Physics, The Chinese University of Hong Kong, Shatin, New Territories, Hong Kong 999077, China

**Keywords:** antigen, nanoparticle delivery system, antigen quantification methodology, antigen release, fluorescence-labeled antigen

## Abstract

Nanoparticles have been identified in numerous studies as effective antigen delivery systems that enhance immune responses. However, it remains unclear whether this enhancement is a result of increased antigen uptake when carried by nanoparticles or the adjuvanticity of the nanoparticle carriers. Consequently, it is important to quantify antigen uptake by dendritic cells in a manner that is free from artifacts in order to analyze the immune response when antigens are carried by nanoparticles. In this study, we demonstrated several scenarios (antigens on nanoparticles or inside cells) that are likely to contribute to the generation of artifacts in conventional fluorescence-based quantification. Furthermore, we developed the necessary assay for accurate uptake quantification. PLGA NPs were selected as the model carrier system to deliver EsxB protein (a *Staphylococcus aureus* antigen) in order to testify to the feasibility of the established method. The results showed that for the same antigen uptake amount, the antigen delivered by PLGA nanoparticles could elicit 3.6 times IL-2 secretion (representative of cellular immune response activation) and 1.5 times IL-12 secretion (representative of DC maturation level) compared with pure antigen feeding. The findings above give direct evidence of the extra adjuvanticity of PLGA nanoparticles, except for their delivery functions. The developed methodology allows for the evaluation of immune cell responses on an antigen uptake basis, thus providing a better understanding of the origin of the adjuvanticity of nanoparticle carriers. Ultimately, this research provides general guidelines for the formulation of nano-vaccines.

## 1. Introduction

Nanoparticles have been widely used as antigen carriers, and many reports have shown an enhanced immune response attributed to the adjuvanticity of nanoparticle carriers [1,2,3,4]. Adjuvanticity is sometimes explained by the increased cellular uptake of antigens when they are carried by nanoparticles, but in most cases, the exact mechanism is unclear [5,6,7]. In fact, quantifying antigen uptake at the cellular level is essential to understanding immune response, including MHCII antigen presentation by immune cells and, consequently, the interpretation of T-cell activation efficiency [8]. Casual or inaccurate quantification of antigen cellular uptake becomes particularly problematic when nanoparticles are used as carriers for antigens, as these carriers can significantly enhance the cellular uptake of antigens through various receptor-mediated or non-specific endocytic processes [9,10,11]. This ambiguity makes it challenging to evaluate the origin of the observed enhancements in immune response and introduces uncertainty in identifying the most effective strategies for vaccine design.

Fluorescence-based techniques that label antigens with fluorescent dyes are commonly used for quantifying antigen uptake [12,13]. These techniques include confocal microscopy, flow cytometry, and fluorescence plate readers [14,15,16,17,18]. Confocal microscopy [12,16,19] allows for the direct visualization of antigen-loaded nanoparticles within cells, enabling differentiation between internalized and membrane-bound nanoparticles. However, it is a low-throughput method, requiring large sample sizes for statistical evaluation [20]. Flow cytometry [17,18] is a more efficient method in terms of throughput, but it cannot distinguish between internalized and surface-bound nanoparticles. This technique is commonly employed for understanding the effect of particle physicochemical parameters on adjuvanticity-based immune responses. However, it can only give information on enhanced antigen internalization by immune cells and increased antibody production [21,22]. The contribution of nanoparticles themselves to immune response activation is unable to be obtained with this method. Fluorescence plate readers provide results similar to those of flow cytometry but are less sensitive [14,23]. One inherent issue with all of these techniques is the fluorescence properties of labeling dyes, as their emission can be affected by chemical and biological environments [23,24,25]. Consequently, under- or over-estimation of antigen quantification is possible in such cases.

The correction of artifacts in common fluorescence-based characterizations is indeed a lab improvement. However, the protocol refers to the two-step procedure of dividing the fed DCs into two groups, one for trustworthy antigen uptake quantification and the other for immune response quantification. These two measurements are then correlated to obtain the immune response on an antigen uptake basis.

In the present study, a model nano-vaccine system needed to be selected to establish the methodology and further validate the effectiveness of the method. PLGA NPs were taken as the model system because of their biodegradability and low systemic toxicity [26]. Additionally, studies have reported the dual functionality of PLGA nanoparticles as both antigen delivery systems and nanoparticles possessing adjuvant properties for enhancing immunity by promoting antigen engulfing by DCs, activating and maturating DCs, and inducing an effective immune response [27]. In terms of antigen selection, we used S. aureus EsxB as a model antigen, one of the important virulence factors secreted by S. aureus [28]. We utilized fluorophore-labeled EsxB, specifically Fluorescein isothiocyanate (FITC-EsxB), and then conjugated it to PLGA NP systems using the conventional amide reaction method [29] as a model system to investigate the quantification of FITC-EsxB loaded in PLGA nanoparticle carriers under different environmental conditions. We observed variations in antigen quantification depending on whether the labeled antigen was located on the surface of the nanoparticles or exposed to specific chemical or biological environments. To address this, we developed a standardized protocol for quantifying antigen uptake based on freely dispersed labeling dyes in an identical medium, subsequent to their release from their original environment. This quantification method enabled us to evaluate immune cell responses on an antigen uptake basis, thereby distinguishing the contributions of increased antigen uptake from other factors related to adjuvanticity in the observed enhanced immune response.

## 2. Materials and Methods

### 2.1. Generation of Bone Marrow-Derived Dendritic Cells (BMDCs)

BALB/c mice were obtained from Air Force Medical University Laboratory Animal Services Centre. All animal experiments were conducted in accordance with the protocols approved by the Air Force Medical University Animal Experimentation Ethics Committee.

The mice were euthanized and then immersed in a 70% ethanol solution for 5 min. The tibias and femurs were cut off, and the muscles were removed under sterile conditions and then soaked in RPMI-1640 medium (Gibco, Grand Island, NY, USA). The ends of the bone were both cut off with scissors, and a 1 mL syringe needle was inserted into the bone cavity to rinse the bone marrow cells from the cavity into a sterile culture dish with RPMI-1640 medium. The cell suspension in the dish was collected and centrifuged at 300× *g* for 5 min, and the supernatant was discarded. The cell pellet was resuspended in Tris-NH_4_Cl red blood cell (RBC) lysis buffer (eBioscience, San Diego, CA, USA); to lyse RBCs. Following the second centrifugation, the supernatant was discarded; then, the pelleted cells were washed with PBS and collected [30]. In the culture process, the cells, suspended in RPMI-1640 medium supplemented with 10% FBS (Gibco, Grand Island, NY, USA), were distributed into 6-well non-treated culture surface plates (Thermo Fisher Scientific, Waltham, MA, USA) at a density of 1 × 10^6^ cell/mL. Subsequently, granulocyte colony-stimulating factor (GM-CSF) (BioLegend, San Diego, CA, USA) was added to the medium at a final concentration of 20 ng/mL. The cells were cultured at 37 °C in an incubator containing 5% CO_2_. The culture medium was completely replaced 3 days later to remove the unattached cells and cell debris. Then, the fresh medium was supplemented with GM-CSF. On day 7, BMDCs, the semi-suspended cells and loosely attached cells, were collected by gently pipetting the medium against the plate, while BMDMs were adherent on the plates.

### 2.2. BMDC Cell Characterization

The examination of BMDC morphologies was conducted using a Nikon A1 Spectral Confocal Microscope (Nikon, Tokyo, Japan). PE-labeled rabbit anti-mouse CD11c, APC-labeled rabbit anti-mouse CD80, and FITC-labeled rabbit anti-mouse MHCII specific antibodies, as well as Hoechst 33342 nuclear stain, were obtained from Biolegend (San Diego, CA, USA). BMDCs were stained with antibodies against CD11c, CD80, and MHCII. The staining process was carried out at 4 °C in PBS with an antibody concentration of 0.1 μg/mL, followed by washing with PBS three times. To stain the BMDC nuclei, the cells were incubated for 10 min at 4 °C in medium containing Hoechst 33342 at a final concentration of 10 μg/mL, following the manufacturer’s protocol. The expression of CD80 and MHCII on BMDCs was evaluated using confocal microscopy, and CD11c expression was detected using flow cytometry. BMDC characterization is shown in Appendix A.

### 2.3. Expression and Purification of EsxB Antigen

EsxB was chosen as the model antigen to investigate the quantification of antigens loaded in nanoparticle carriers under different environmental conditions. His-tagged EsxB was prepared using the standard IPTG (Merck, Shanghai, China) induction protocol [31]. Briefly, full-length wild-type EsxB DNA was subcloned into the Pet-28a vector and expressed as a 6 × His-tage fusion protein in BL21 (DE3) E. coli (TIANGEN BIOTECH (BEIJING) Co. Ltd, Beijing, China). The expression was induced using IPTG; purified from bacterial cell lysates by Ni-Sepharose (Cytiva, Shanghai, China); washed 3 times in lysis buffer; and gradient-eluted with phosphate buffer saline (PBS), and 50 mmol/L, 100 mmol/L, 300 mmol/L, and 500 mmol/L imidazole (Merck, Shanghai, China). The sample was then dialyzed with PBS, and endotoxin was removed with a High-Capacity Endotoxin Removal column (Xiamen Bioendo Technology Co. Ltd, Xiamen, China). Finally, the EsxB solution was filtered through a 0.22 μm sterile filter, and the total protein concentration was quantified using a bicinchoninic acid (BCA) protein assay (Thermo Fisher Scientific, Waltham, MA, USA) against bovine serum standard. The purity and specificity of the EsxB antigen were analyzed using SDS-PAGE (Appendix A).

### 2.4. Fluorescence Labeling of Protein Antigens

To prepare fluorescein isocyanate-labeled EsxB, 1 mg of EsxB was dissolved in 9 mL of a Sodium Bicarbonate–Sodium Carbonate (NaHCO_3_/Na_2_CO_3_) buffer solution (pH 9). FITC (Thermo Fisher Scientific, Waltham, MA, USA) in DMSO solution (2 mg/mL) was then added to the EsxB solution for incubation at 4 °C overnight. After that, 1.4 mL of 1 M HCl was added to neutralize the reaction mixture. The mixture was concentrated to 2 mL using a 10 kDa cutoff centrifugal concentrator (Merck Millipore, Billerica, MA, USA);. The concentrated solution was then dialyzed to remove unreacted FITC. The final concentration of FITC-EsxB was determined with the BCA method (Appendix A).

### 2.5. Preparation and Characterization of FITC-EsxB-PLGA Nanoparticles

FITC-EsxB-PLGA NPs were prepared with the emulsion–solvent evaporation method [3,32]. Briefly, 100 mg PLGA (Xi’an Ruixi Biological Technology Co. Ltd, Xi’an, China) was dissolved in 2.5 mL mixed solvents (DCM: Acetone = 3:2, *v*/*v*). Then, the PLGA organic phase was emulsified into 10 mL of a 5% PVA (Mw 31,000 Da–50,000 Da, 87–89% hydrolyzed) (Merck Millipore, Billerica, MA, USA) water phase using sonication (2 min, 20 W) until a fine emulsion was formed. The formed emulsion was added to 50 mL of deionized water and stirred for 4 h at room temperature to evaporate DCM and acetone. After that, the NPs were collected and washed three times using centrifugation at 16,500× *g* at 4 °C and then passed through a 0.45 μm filter, resulting in sphere-shaped PLGA NPs with diameter of 170 nm. The NPs were freeze-dried and stored at −20 °C for later use.

FITC-EsxB was chemically conjugated to the PLGA NPs through a condensation reaction between the carboxyl group on the NPs and the amino group of EsxB. To achieve this, 10 mg of PLGA NPs was dispersed in 10 mL of 25 mmol/L MES buffer, and 0.4 mL of 1 mol/L EDC and 0.25 mL of 1 mol/L NHS (Merck Millipore, Billerica, MA, USA) were added to the mixture and stirred for 4 h at room temperature to activate the carboxyl groups on the NPs. After that, the NPs were collected and washed three times using centrifugation at 16,500× *g* at 4 °C. The activated NPs were suspended in 1 mL of 1 mg/mL EsxB solution overnight at 4 °C; then, the FITC-EsxB-PLGA NPs were collected and washed three times using centrifugation at 16,500× *g* at 4 °C resulting in ~180 nm diameter sphere-shaped FITC-EsxB-PLGA NPs. The nanoparticles were freeze-dried and stored at −20 °C for later use. 

The characterization of the FITC-EsxB-PLGA NPs is depicted in Appendix A and Appendix A. The size distribution, zeta potential, and polydispersity index (PdI) of the NPs were measured using a Zetasizer PRO (Malvern, Worcestershire, UK). The successful binding of EsxB to the NPs was observed with FTIR spectra (Thermo scientific Quattro S; Waltham, MA, USA) from samples of EsxB, PLGA NPs, and FITC-EsxB-PLGA NPs. The encapsulation efficiency (EE) and loading capacity (LC) of FITC-EsxB on the NPs were determined using a BCA protein quantification kit (Beyotime, Shanghai, China). FITC-EsxB was isolated from the NPs by dissolving the NPs in DMSO/PBS solution. The EE (%) and LC (%) can be calculated as follows [3]:EE%=WFITC-EsxB on the NPs (mg)WFITC-EsxB input (mg)×100%
LC%=wFITC-EsxB on the NPs (mg)WFITC-EsxB-PLGA NPs (mg)×100%
where *W_FITC-EsxB on the NPs_* is the total amount of FITC-EsxB on the NPs determined with the BCA method, *W_FITC-EsxB input_* is the total amount of the initial feeding FITC-EsxB, and *W_FITC-EsxB-PLGA NPs_* is the amount of total FITC-EsxB-PLGA NPs.

### 2.6. Biocompatibility Assessment of FITC-EsxB-PLGA NPs In Vitro Using BMDCs

Cell viability was evaluated using a colorimetric cell counting kit-8 (CCK-8) by Beyotime. DCs were plated in 96-well plates at a density of 1 × 10^5^ cells/mL in 150 μL of growth medium per well. The wells were then treated with 50 μL of EsxB-PLGA NPs at different NPs concentrations prepared by diluting a stock solution in growth medium. After 24 h, 10 μL of CCK-8 reagent was added to each well and incubated at 37 °C for 4 h. Absorbance at 450 nm was measured, and the results were calculated based on the ratio of the average O.D. 450 values of wells containing NP-stimulated cells to those containing only cells with medium. The cell viability result is shown in Appendix A.

### 2.7. Immunization of Mice and Collection of EsxB-Specific Spleen Cells

Female BALB/c mice, aged 6–8 weeks, were randomly divided into two groups (*n* = 6) and subcutaneously immunized with EsxB-PLGA NPs (50 µg/mouse) or PBS (as a negative control) on day 0. (The NPs were suspended in PBS and sterilized before administration). Boost immunizations were administered on day 14 and 28 with a reduced dose of 25 µg/mouse. After 35 days from the initial vaccination, ELISA was performed to determine the EsxB-specific antibody titers in the sera using standard protocols [33]. Briefly, blood samples were collected via the tail vein to prepare the serum samples. An ELISA plate (Corning, New York, USA) was coated with 500 ng/well of EsxB diluted in a medium containing 50 mmol/L carbonate buffer (pH 9.6) and then incubated overnight at 4 °C. The plate was then washed five times with PBST (0.5% Tween 20). Serum samples (100 μL/well) were added to the ELISA plate with serial dilutions ranging from 1:200 to 1:204,800 and incubated at 37 °C for 1 h. The plate was washed with PBST five times. HRP-labeled goat anti-mouse IgG (1:5000, 100 μL/well) was added and incubated at 37 °C for 45 min. After discarding the IgG medium, the plate was washed with PBST five times. ABTS substrate chromogen (Merck, Shanghai, China) was added to each well, and the final color development occurred for 30 min at room temperature in the dark. The absorbance was measured at 405 nm using a microplate reader (BioTek, Winooski, VT, USA). The EsxB-specific antibody titers were presented as the mean serum ELISA area under the curve (AUC) with standard deviation (SD) for the six mice. The cutoff value was calculated as an O.D. value above 0.1 (Appendix A).

EsxB-specific spleen cells were harvested from EsxB-immunized mice following standard protocols [34]. Briefly, the spleens were aseptically separated from the euthanized EsxB-immunized mice and washed three times with PBS buffer. The spleens were then crushed and pressed through a 200-mesh steel mesh to obtain a homogeneous cell suspension. After centrifugation at 1500× *g* rpm for 5 min, red blood cells were removed using a Tris-NH_4_Cl red blood cell (RBC) lysis buffer (pH 7.2). The cell pellets were resuspended in RPMI-1640 medium, diluted to a concentration of 5 × 10^6^ cells/mL, and incubated at 37 °C in a humidified incubator with 5% CO_2_ for further experiments. 

### 2.8. Accurate Quantification of Antigen Amount in Different Environments

The fluorescence properties of FITC and FITC-EsxB in different environments were examined with fluorescent spectra using a Synergy HTX microplate reader (BioTek, Winooski, VT, USA). The fluorescence intensity of FITC at a concentration of 5 µg/mL was measured in different solvents (PBS buffer, 10% SDS buffer, and PBS/DMSO buffer (1:2, *v*/*v*)) with excitation at 490 nm and emission detection at 525 nm. To determine the concentrations of FITC-EsxB in specific solvent, calibration curves of FITC-EsxB in different solvents were established, respectively (Appendix A). 

The fluorescence of free FITC-EsxB in/released from endosomes was then examined using the same fluorescence method as described above. In detail, DCs were seeded in 96 well plates (tissue culture-treated, *n* = 6) at a density of 1 × 10^5^ cells/well. FITC-EsxB (1 mg/mL) was incubated with the cells overnight. We centrifuged the plate (300× *g*, 5 min), discarded the supernatant, and washed the cells with PBS 3 times. Three wells of the treated cells were resuspended in PBS, while the other three wells of cells were lysed with 10% SDS for 1 h. A fluorescence microplate reader was then employed to read the fluorescence intensity of FITC in each well (λ_ex_ = 490 nm/λ_em_ = 525 nm). The total protein concentration of each well was then determined using the calibration curve established above. 

The fluorescence of FITC-EsxB absorbed on the surface of PLGA NPs and those released from the nanoparticle surface by PBS/DMSO buffer was also determined using fluorescent spectrum analysis. Briefly, equal amounts of FITC-EsxB-PLGA NPs (200 μg/mL) were dissolved in PBS and PBS/DMSO, respectively. The EsxB concentration in each well was determined using the same protocol as the one for free FITC-EsxB in/released from cells. 

### 2.9. Quantification of FITC-EsxB Cellular Uptake

In order to investigate FITC-EsxB cellular uptake, the fluorescence method was also employed to measure the intracellular antigen amount quantitatively. DCs were seeded into 96-well plates at a cell density of 3 × 10^5^ /well. The serial concentrations of FITC-EsxB and FITC-EsxB-PLGA NPs were then added to DCs for overnight incubation at 37 °C. Then, antigen-containing medium was thoroughly washed away with PBS. Cells were then lysed with 10% SDS and NPs dissolved in DMSO for further fluorescence investigation. A fluorescence microplate reader was then employed to read the fluorescence intensity of FITC in each well (λ_ex_ = 490 nm/λ_em_ = 525 nm). The total antigen uptake amount was determined using the calibration curve established in 10% SDS or DMSO. FITC-EsxB cellular uptake (pg/DC) was finally established as a function of FITC-EsxB feeding amount (pg/DC) (Appendix A). 

### 2.10. Determination of IL-2 Secretion by EsxB-Specific Spleen Cells

The presentation of EsxB-specific peptide after antigen processing by DCs on class II major histocompatibility complex (MHCII) was assessed according to IL-2-secreting EsxB-specific T cells. DCs were seeded at a density of 3 × 10^5^ cells per well in a 96-well plate and incubated with antigen, in nanoparticle formulation or freely dissolved, overnight at 37 °C. Each antigen sample was added to 6 wells of DCs. The medium containing the antigen was thoroughly washed away with PBS. Antigen uptake was determined in 3 wells for each sample. Antigen cellular feeding was calculated by dividing the total antigen feeding amount by the total number of DCs. For the remaining 3 wells of each sample, 200 μL of EsxB-specific spleen cells at 6 × 10^6^ cells in RPMI-1640 full medium was added and incubated overnight. The culture medium was collected after centrifugation at 500× *g* for 5 min. The IL-2 level in culture medium was assayed using a Mouse IL-2 ELISA kit (Abcam, Cambridge, UK). In the data analysis, the IL-2 secretion amount per EsxB-specific T cell was calculated in two steps. First, the IL-2 secretion amount per spleen cell was calculated by dividing the total IL-2 secretion amount by the total number of spleen cells; then, according to the proportion of antigen-specific T cells in the spleen cells (0.02), the IL-2 secretion amount per EsxB-specific T cell was calculated by dividing the IL-2 secretion amount per spleen cell by the coefficient 0.02.

### 2.11. The Expression of CD86 and MHCII on BMDCs

In order to investigate the influence of NPs on DC maturation, DCs were seeded at 3 × 10^5^ cells per well in a 6-well plate and treated with EsxB, PLGA NPs, EsxB-PLGA NPs, or PBS (as the control group) at a concentration of 100 μg/mL for overnight incubation at 37 °C. Cells were then washed with PBS and labeled with APC-labeled rabbit anti-mouse CD86 and FITC-labeled rabbit anti-mouse MHCII specific antibodies. The expression of CD86 and MHCII on BMDCs was measured using flow cytometry. Flow cytometry was conducted on an ACEA Novo Express system (Agilent Bio, Santa Clara, CA, USA), and the data were analyzed using FlowJo software (TreeStar, San Francisco, CA, USA). The results were expressed as percentages of positive cells.

### 2.12. The Measurement of IL-12 Secretion by DCs

DCs were co-cultured with FITC-EsxB, FITC-PLGA NPs, and FITC-EsxB-PLGA-NPs at a density of 3 × 10^5^ cells/well for 12 h. Cells were then centrifuged, and the supernatants were collected to quantify the IL-12 secretion amount by DCs using a Mouse IL-12 ELISA Kit (Abcam, Cambridge, UK). The IL-12 secretion amount per cell was calculated by dividing total IL-12 secretion by the total number of DCs.

## 3. Results

### 3.1. The Influence of Different Chemical Environments on FITC Fluorescence Intensity

Figure 1 in the study compares the fluorescence signal of FITC, a commonly used fluorophore label, in different chemical environments, including PBS, 10% SDS, and PBS/DMSO buffer (1:2 *v*/*v*). Despite the use of the same concentration of FITC (5 μg/mL), the fluorescent intensities were significantly different. This discrepancy arises from the fact that various parameters can affect the quantitative emission of fluorescent dyes, such as salt concentration, the presence of detergents, and the hydrophobicity/hydrophilicity of the solvent [35,36]. The surrounding medium can impact the energy, shape, and intensity of the absorption/emission spectra of a fluorescent dye [23]. FITC, as a hydrophobic molecule, exhibits high fluorescence in organic media like PBS/DMSO. However, in aqueous environments like PBS, FITC is prone to quenching due to aggregation [37]. This observation aligns with the general rule in fluorophore calibration, which states that the fluorophore concentration calibration for fluorescence labeling-based antigen quantification should be conducted in the same medium as the sample. However, fulfilling this requirement can be challenging in in vitro quantification, which is commonly found in the existing literature [38,39,40].

### 3.2. The Influence of Different Chemical/Biological Environments on the Fluorescence Quantification of FITC-Labeled Antigen (EsxB)

We further tested this hypothesis (inaccurate antigen quantification in vitro) by labeling a model antigen (EsxB) with FITC and subjecting it to different chemical and biological environments. In the first set of experiments, we dissolved free FITC-EsxB in different chemical solvents, namely, PBS, 10% SDS, and PBS/DMSO buffer. Despite using the same concentration of FITC-EsxB, we observed significant differences in the fluorescence intensity among these solvents (Figure 2a). This finding is consistent with the observations from Figure 1, where only free FITC was used. In the second set of experiments, we compared the fluorescence intensity of FITC-EsxB that was endocytosed by BMDCs (Appendix A) in PBS buffer with that released from BMDCs and dispersed into 10% SDS buffer. Once again, despite using the same feeding concentration of FITC-EsxB, we observed significant differences in the fluorescence intensity between these two groups (Figure 2b). In the third set of experiments, we compared the fluorescence intensity of FITC when FITC-EsxB was conjugated on the surface of PLGA nanoparticles (FITC-EsxB-PLGA NPs) with that when it was released from the NPs and dispersed in PBS/DMSO buffer. Similar to the previous experiments, in this experiment, we found a significant increase in the fluorescence intensity of FITC-EsxB when it was released by DMSO compared with that on the nanoparticle surface (Figure 2c).

Quantification of FITC-EsxB in three different chemical environments (PBS, 10% SDS, PBS/DMSO) was then carried out by establishing standard calibration using the respective solvents (Appendix A). We then compared the FITC-EsxB quantification in these environments to the BCA method, which is a standard protein quantification method (Appendix A). As expected, when the right calibration was used (i.e., the identical solvent as the sample), the chemical environment no longer affected protein quantification. We found similar FITC-EsxB concentrations in the three chemical environments (PBS, 10% SDS, PBS/DMSO mixed solution) (Figure 3a).

However, when FITC-EsxB was endocytosed by dendritic cells or conjugated on the surface of PLGA nanoparticles (FITC-EsxB-PLGA NPs), accurate quantification became difficult. This was due to the lack of information about the exact chemical/biological environment of the FITC-EsxB dye. In these cases, it was not possible to establish the right in situ calibration. Consequently, differences in antigen quantification were expected when FITC-EsxB was released from specific chemical environments (e.g., endosomes or nanoparticles) and dispersed into a known medium (Figure 3b,c). These findings further highlight the challenges in accurately quantifying FITC-labeled antigens when they are internalized by cells or conjugated to nanoparticles. The lack of control over the specific chemical/biological environment of the dye makes it difficult to establish appropriate calibration. Therefore, accurate quantification may be compromised in such scenarios.

The results obtained suggest a potential solution for accurate quantification of antigens when they are loaded onto nanoparticle carriers or taken up by antigen presentation cells. It is essential to develop a protocol that ensures the release of fluorophore-labeled antigens from their local environment and disperses them into a standardized medium before quantification. This solution is supported by the experimental results shown in Figure 3c. In this experiment, similar FITC-EsxB quantification was achieved using the releasing strategy (middle column) or BCA analysis (right column) when the same FITC-EsxB-conjugated PLGA nanoparticles were employed. This indicates that by releasing the antigens from their specific chemical environments and dispersing them into a known medium, accurate quantification becomes feasible. Additionally, this approach allows for the establishment of a correlation between antigen uptake and the amount of antigen fed per dendritic cell.

### 3.3. The Quantification Methodology for Antigen Uptake and IL-2 Secretion (Represented for T-Cell Activation) at the Cellular Level

The accurate quantification of antigens loaded onto nanoparticle carriers enables the evaluation of immune cell responses based on the uptake of antigens per cell. This can be achieved by dividing the DCs into two identical batches (Figure 4). The first batch is used to quantify antigen uptake, as explained earlier, while the second batch is used for immune response analysis (we used IL-2 secretion as an example). IL-2 secretion was analyzed by employing DC-EsxB-spleen cells obtained from mice immunized with EsxB. This experimental setup allowed us to assess IL-2 secretion by T cells as a function of the amount of antigen absorbed per DC. Consequently, we obtained a relationship between IL-2 secretion per T-cell and antigen uptake per DC cell (Figure 4).

Using the developed method for antigen quantification, our initial investigation focused on determining whether the PLGA nanoparticle carrier possesses adjuvanticity and whether this adjuvanticity is solely attributed to the enhanced cellular uptake of antigens when loaded onto PLGA nanoparticles.

We initially established a correlation between the amount of antigen uptake and the amount of antigen fed per DC. By examining the triplicated results presented in Appendix A, we observed a linear relationship between the amount of antigen fed and the amount of antigen uptake for both the free FITC-EsxB and FITC-EsxB-PLGA NP groups per cell. Furthermore, our findings demonstrate that the use of PLGA nanoparticles as antigen carriers significantly enhances the cellular uptake of antigens, even with a small amount of antigen feeding. These results are consistent with previous literature reports [41].

Furthermore, we conducted the quantification of IL-2 secretion by EsxB-specific T cells as a function of antigen feeding amount per cell (Appendix A). We observed a linear dependence between IL-2 secretion by EsxB-specific T cells and FITC-EsxB feeding amount. Notably, when PLGA NPs were used as the carrier for the antigen, enhanced IL-2 secretion was observed.

Next, we plotted the fitting curves of IL-2 secretion as a function of FITC-EsxB uptake at the same EsxB-FITC feeding amount. From Appendix A, both free antigen and FITC-EsxB-PLGA NPs exhibited linear relationships between antigen uptake and IL-2 secretion by EsxB-specific T cells. Importantly, FITC-EsxB-PLGA NPs induced 3–5 times stronger IL-2 secretion compared with free antigen (Figure 5). Given that IL-2 secretion reflects MHCII presentation efficiency, these results strongly suggest that PLGA NPs enhance the immune response when employed as antigen carriers.

### 3.4. The Quantification of Antigen Uptake and IL-12 Secretion (Represented for DC Maturation) at the Cellular Level Based on the Established Method

We here confirmed that the presence of PLGA nanoparticle carriers promotes the immune response of the respective immune cells, and this improvement is not solely attributed to the enhanced cellular uptake of the antigen when loaded onto the nanoparticles. Given that dendritic-cell maturation is a crucial step in the immunization process, we initially examined the maturation of DCs. As shown in Figure 6a, the upregulation of MHCII and CD86 serves as an important indicator of DC maturation [42,43]. Additionally, the secretion of IL-12 by DCs also influences the expression of MHCII and CD86, further facilitating the maturation of DCs [44].

We subsequently investigated the secretion of IL-12 by DCs. We compared the effects of pure FITC-PLGA NPs, FITC-EsxB-PLGA NPs, and a PBS group as the negative control. As depicted in Figure 6b, both the blank nanoparticles and FITC-EsxB-PLGA NPs induced significantly higher IL-12 secretion compared with the PBS group. Moreover, the FITC-EsxB-PLGA NPs exhibited a greater capacity to promote IL-12 secretion compared with the pure nanoparticles. Next, we examined the expression of MHCII and CD86 on the surface of DCs after treating them with pure PLGA NPs and EsxB-PLGA NPs. It was observed that EsxB-PLGA NPs significantly upregulated the expression of both MHCII and CD86 on the surface of DCs in comparison to the PBS group. Pure PLGA NPs also significantly enhanced the expression of CD86 (Figure 6c and Appendix A) but not MHCII. These findings suggest that both EsxB-PLGA NPs and pure PLGA NPs can stimulate the maturation of DCs, as evidenced by the upregulation of MHCII and CD86. Furthermore, the EsxB-PLGA NPs exhibited an additional effect of promoting IL-12 secretion by DCs, indicating their potential to enhance the immune response.

## 4. Discussion

Nanoparticles serving as antigen delivery systems or vaccine adjuvants have been well studied in recent years [45]. Firstly, nanoparticles as antigen delivery systems can significantly increase antigen uptake, which, in turn, increases the amounts of MHCII-antigen peptides presented on the APC surface, leading to a stronger adaptive immune response [45]. On the other hand, nanoparticles perform as vaccine adjuvants due to their biophysical and biochemical properties that enhance antigen uptake, processing, and presentation. 

Identifying the origin of nanoparticles’ adjuvanticity is crucial to designing a specific nano-vaccine system with a potent immune response. In this regard, differentiating the contribution of enhanced antigen uptake from other causes is needed. While measuring the fluorescence signal from intracellular dye-labeled antigens is a commonly adopted method to estimate antigen uptake [46], the complexity of the intracellular environment makes it unlikely to calibrate the measured intensity to the actual uptake quantity, leading to inaccurate results [23]. This hypothesis is proven by our experimental results of FITC-labeled EsxB antigen located in cells or on PLGA nanoparticles: direct fluorescence measurements failed to disclose the actual quantity of EsxB antigen due to the absence of reliable calibration.

The above understanding leads to a simple solution that can quantify the actual amount of antigen absorbed, that is, releasing the antigen (that has been trapped in the specific physical/biochemical environment) into a known chemical solution using the standard calibration method. Our experimental results shown in Figure 3 demonstrate the effectiveness of such a method. 

Once quantification can be carried out, it is possible to analyze the response of specific immune cells on an antigen uptake basis and thus exclude the contribution from enhanced antigen uptake. In other words, the promoted immune response must originate from more effective antigen processing and presentation. The feasibility of this newly developed protocol was demonstrated when we characterized the response of EsxB-specific T cells (using IL-2 secretion) as a function of EsxB antigen uptake. By comparing results shown in Figure 5 and Appendix A, one can clearly see the contribution of enhanced antigen uptake (when loaded in PLGA NPs vs. free EsxB antigen) to the increased IL-2 secretion, which is consistent with most of the literature observations. More importantly, the data also revealed unambiguous adjuvanticity of the EsxB-loaded PLGA NPs irrelevant to antigen uptake. A further understanding of adjuvanticity comes from the analysis of DC maturation (characterization of MHCII and co-stimulatory molecules of CD86, and cytokine IL-12) upon antigen uptake: the EsxB-loaded PLGA NPs were found to promote DC maturation and therein EsxB-specific T-cell response.

Based on the research results in the present research, an accurate quantification methodology to study the antigen uptake and nanoparticles adjuvanticity has been well established. Since PLGA NPs are the model system we chose to deliver the model antigen, EsxB, not a specific choice, the established methodology could also be applied with other antigen carriers, such as liposomes, micelles, exosomes, and so on. The method provides a new tool for vaccine adjuvant selection and paves a new way for nano-vaccine design.

## 5. Conclusions

The quantification of antigen uptake per cell relies on the release of fluorescent-labeled antigens from a specific environment into a standard one. In this study, we have successfully developed a method to accurately quantify antigen uptake per cell. A linear relationship between antigen uptake and IL-2 secretion per cell was identified. According to the established method, we found that for the same antigen uptake amount, the antigen delivered by PLGA nanoparticles could elicit 3.6 times IL-2 secretion (representative of cellular immune response activation) and 1.5 times IL-12 secretion (representative of DC maturation level) compared with pure antigen feeding. Thus, by utilizing this method, we can easily discern whether the enhanced immune response is primarily due to increased antigen uptake or the adjuvanticity of the nanocarriers This quantification method adds to the existing tools for evaluating the carrier or adjuvant function of nanoparticles, opening up new possibilities for screening effective antigen delivery systems or vaccine adjuvants.

## Figures and Tables

**Figure 1 vaccines-12-00028-f001:**
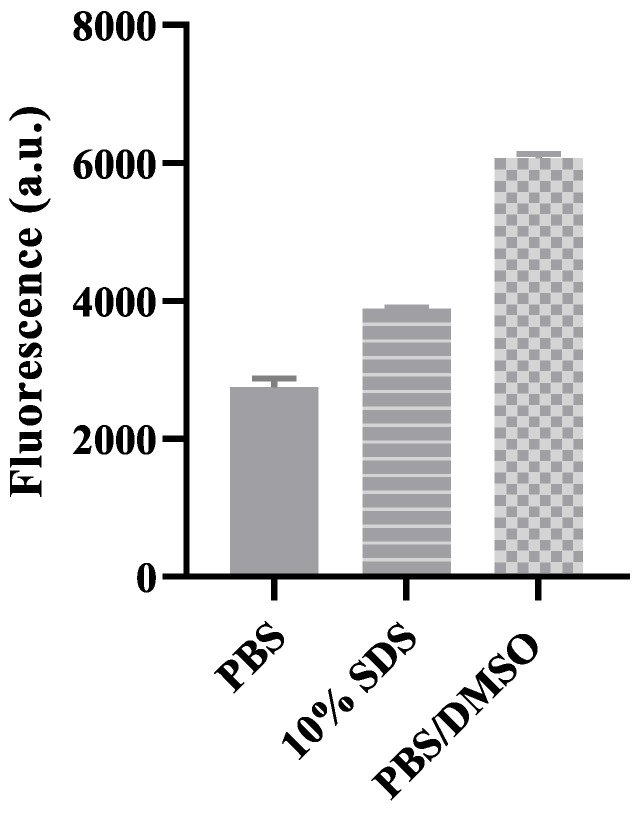
The intensity of FITC in three different solvents.

**Figure 2 vaccines-12-00028-f002:**
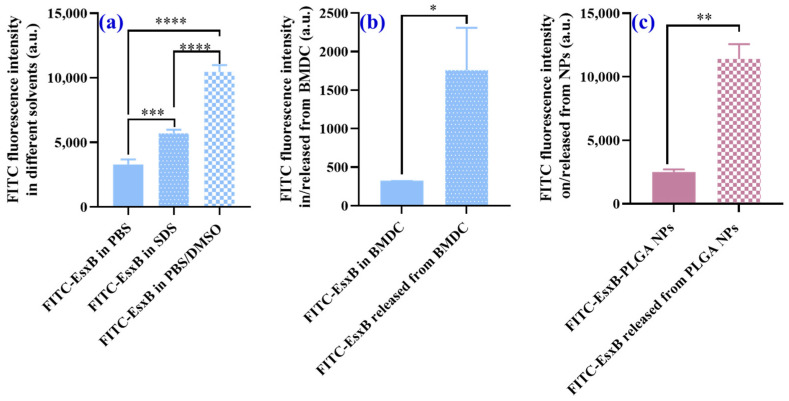
Fluorescence intensity of free FITC-EsxB in different chemical/biological environments: (**a**) in different solvents (PBS, 10% SDS, PBS/DMSO buffer), the input FITC-EsxB concentration was 5 μg/mL; (**b**) in BMDCs or released from BMDCs (after cell incubation with FITC-EsxB overnight and lysed by 10% SDS for 1 h), co-cultured FITC-EsxB was 1 mg/mL; and (**c**) on PLGA NPs or released from NPs (in PBS/DMSO buffer for 1 h), the same PLGA NPs (with FITC-EsxB conjugated on the surface) (200 μg/mL) were used in the experiments. (n = 3; * *p* < 0.05, ** *p* < 0.01, *** *p* < 0.001, **** *p* < 0.0001, analyzed using one-way ANOVA and *t*-test.)

**Figure 3 vaccines-12-00028-f003:**
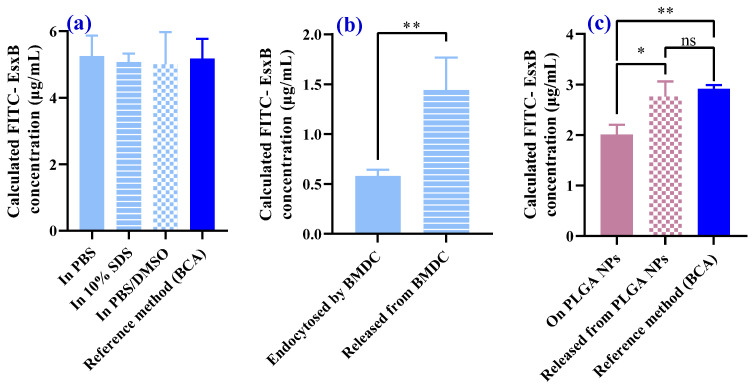
FITC-EsxB quantification in different chemical/biological environments. (**a**) The same amount of FITC-EsxB was dissolved in three different solvents: PBS, 10% SDS, and PBS/DMSO. The last column is the FITC-EsxB quantification with the standard BCA method. In all four data sets, the input FITC-EsxB concentration was 5 μg/mL. (**b**) FITC-EsxB quantification when endocytosed by BMDCs (1 × 10^5^ cells/well) or released by 10% SDS buffer. (**c**) Quantification of FITC-EsxB conjugated on PLGA NPs in PBS buffer or released from NPs by PBS/DMSO buffer. The last column is the quantification of FITC-EsxB with the standard BCA method. The same PLGA NPs (with FITC-EsxB conjugated on the surface) (200 μg/mL) were used in all three sets of experiments. (*n* = 3; * *p* < 0.05, ** *p* < 0.01, ns: no significant differences, analyzed by one-way ANOVA and *t*-test.)

**Figure 4 vaccines-12-00028-f004:**
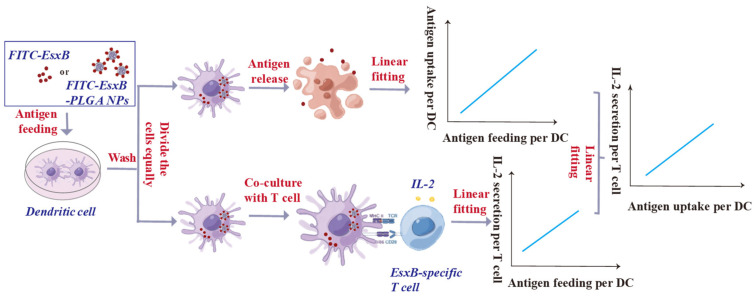
Schematic for the quantification methodology of antigen uptake and IL-2 secretion per cell.

**Figure 5 vaccines-12-00028-f005:**
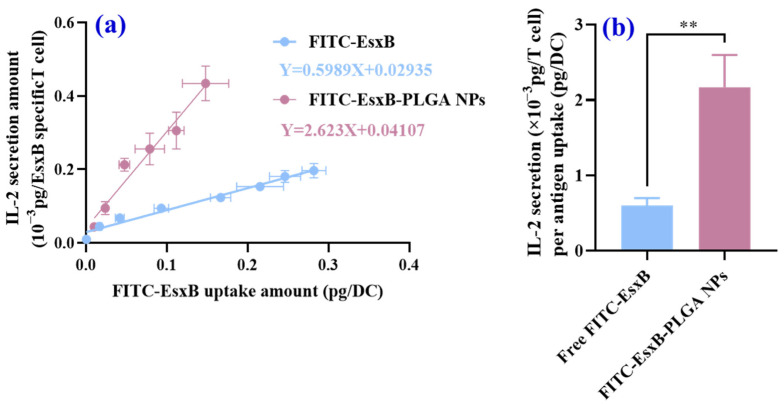
Quantification of IL-2 secretion by EsxB specific T cells. (**a**) Quantification of IL-2 secretion as a function of antigen uptake per cell in free antigen and antigen-NP treatment groups. (**b**) Comparison of slopes of (Appendix A). (*n* = 3, analyzed using one-way ANOVA and *t*-test, ** *p* < 0.01.)

**Figure 6 vaccines-12-00028-f006:**
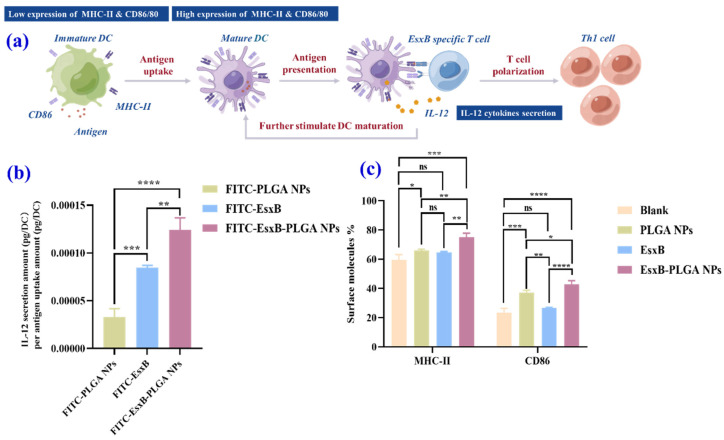
(**a**) Schematic of DC maturation and the process of antigen presentation. (**b**) IL-12 secretion amount by antigen uptake after DCs were exposed to FITC-EsxB, FITC-PLGA NPs, and FITC-EsxB-PLGA NPs. (**c**) Quantitative assay of MHCII/CD86 expression on DCs after they were exposed to PBS, PLGA NPs, EsxB, and EsxB-PLGA NPs using FCM. (*n* = 3, analyzed using one-way ANOVA, * *p* < 0.05, ** *p* < 0.01, *** *p* < 0.001. **** *p* < 0.0001, ns: no significant differences)

## Data Availability

Data will be made available on request.

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
