# Peer review of "Uptake Quantification of Antigen Carried by Nanoparticles and Its Impact on Carrier Adjuvanticity Evaluation"

_vaccines, 2023, doi:10.3390/vaccines12010028_

Round 1
Reviewer 1 Report
Comments and Suggestions for Authors
The manuscript (vaccines-2662458) entitled "Uptake quantification of antigen carried by nanoparticles and its impact on carrier adjuvanticity evaluation" provides timely and interesting information for the reader but needs to be revised considering the following suggestions for further improvement:
1. Title may revised as "Uptake quantification of antigen carried by PLGA nanoparticles and its impact on carrier adjuvanticity evaluation"
2. The abstract needs to be revised to incorporate the absolute results of the findings. More generalized findings already present in the abstract should be deleted/concised and more specific findings including novelty should also be incorporated to make it more interesting for the reader.
3. It is suggested to include previous important findings of similar lines of research particularly on PLGA nanoparticles in the introduction section and highlight the current research gap in this area. How the present investigation is helpful in addressing this research gap or providing value addition in this area of research should be added in the revision. It should be reorganized in a better way to make it more interesting for the reader. Various similar lines of research work have already been investigated by different researchers which should be included in the introduction.
4. The rationale for the selection of PLGA as a polymer for synthesis nanoparticles in the current investigation should be included. It should highlight the reason for PLGA selection comparison to other biodegradable polymers in the introduction section of the revised manuscript.
5. Sub-section of material and method section should be 2.1, 2.2, 2.3, ..... and so on.
6. The title of the sub-section "Particle synthesis" should be "2.5. Preparation and characterization of PLGA nanoparticles"
7. It is suggested "Results" section should be divided into different sub-sections to make it more interesting and quick to understand for the reader.
8. Results of the "Particle synthesis" section (figure S3 and Table 1) should be included in the main manuscript. It should be deleted from supplementary information (figure S3 and Table 1) and included in the main manuscript (figure S3 and Table 1).
9. It is important to include the value of the polydispersity index (PDI) while measuring the hydrodynamic diameter of (i) PLGA NPs and (ii) FITC-EsxB-PLGA NPs.
10. It is very important to include the "level of significance; p-value" in the results of supplementary table 1.
11. The results of hydrodynamic diameter and zeta potential for FITC-EsxB-PLGA NPs in supplementary Figure S3 and supplementary Table 1 are different. Justify this difference.
12. It is important to elaborate on the discussion particularly related to the use of PLGA nanoparticles in vaccine delivery and related challenges and safety concerns. Also, incorporate the reason for low encapsulation efficiency (EF% - 30.7%) and loading capacity (LC% - 3.3%)in the current investigation and elaborate related discussion in the revised manuscript.
Comments on the Quality of English Language
Moderate editing of English language required.
Author Response
Response letter
Dear Editor,
We now received the reviewers’ comments on our manuscript entitle “Uptake quantification of antigen carried by nanoparticles and its impact on carrier adjuvanticity evaluation” (vaccines-2662458). We are grateful to reviewers’ constructive suggestions, which help us to improve the manuscript. We have carefully revised the manuscript based on their comments and a list of changes made can be found in the following. All changes have been marked in the revised manuscript.
Response to the reviewers’ comments:
Reviewer #1
The manuscript (vaccines-2662458) entitled "Uptake quantification of antigen carried by nanoparticles and its impact on carrier adjuvanticity evaluation" provides timely and interesting information for the reader but needs to be revised considering the following suggestions for further improvement:
- Title may revised as "Uptake quantification of antigen carried by PLGA nanoparticles and its impact on carrier adjuvanticity evaluation"
Response to Question 1 of Reviewer #1
Thanks for the reviewer’s comments. Actually, we insisted on keeping the original title because the established method is not only suitable for PLGA nanoparticles as antigen carriers but also applicable for other nanoparticle carriers. To clarify, we insert in the abstract part description for the NPs system and antigen choices as follows,
“PLGA NPs was selected as a model carrier system delivering EsxB protein (S. aureus antigen) in order to testify the feasibility of the established method”. See Page 1, Line 22-23, which were marked in red.
In addition, in the introduction part, we clarified the reason why we chose PLGA and EsxB as model delivery system and antigen in details. The description is as follows, “PLGA NPs was taken as the model system because of its biodegradability and low systemic toxicity [26]. Besides, studies have reported dual functionality of PLGA nanoparticles as both antigen delivery and adjuvant properties for enhancing immunity by promoting antigen engulfing by dendritic cells (DCs), activating and maturating DCs, and inducing effective immune response [27]. In terms of antigen selection, we used S. aureus EsxB as model antigen, one of the important S. aureus-secreted virulence factors [28]. We utilized a fluorophore-labelled EsxB, specifically Fluorescein isothiocyanate (FITC-EsxB), conjugating on PLGA NPs systems by conventional amide reaction method [29], as a model system to investigate the quantification of FITC-EsxB loaded in PLGA nanoparticle carriers under different environmental conditions.”. See Page 2, Line 71-82, which were marked in red.
- The abstract needs to be revised to incorporate the absolute results of the findings. More generalized findings already present in the abstract should be deleted/concised and more specific findings including novelty should also be incorporated to make it more interesting for the reader.
Response to Question 2 of Reviewer #1
The abstract has been revised and more details of the results have been added in the revised version, see Page 1, Line 19 and Line 22-28, which were marked in red.
- It is suggested to include previous important findings of similar lines of research particularly on PLGA nanoparticles in the introduction section and highlight the current research gap in this area. How the present investigation is helpful in addressing this research gap or providing value addition in this area of research should be added in the revision. It should be reorganized in a better way to make it more interesting for the reader. Various similar lines of research work have already been investigated by different researchers which should be included in the introduction.
Response to Question 3 of Reviewer #1
Examples of nano-vaccines were introduced in the revised introduction, in order to illustrate that fluorescence-based technics could only give the information of enhanced antigen uptake when delivered by nanoparticles and finally increased the antibody production in mice. But with similar amount of antigen uptake, not all the cases when antigen delivered by nanoparticles could enhance the immune response than free antigen feeding, meaning that nanoparticles sometimes only served as antigen carrier not adjuvant. See Page 2, line 60 which were marked in red.
- The rationale for the selection of PLGA as a polymer for synthesis nanoparticles in the current investigation should be included. It should highlight the reason for PLGA selection comparison to other biodegradable polymers in the introduction section of the revised manuscript.
Response to Question 4 of Reviewer #1
The rationale for the selection of PLGA has been added in the introduction as follows, “PLGA NPs was taken as the model system because of its biodegradability and low systemic toxicity. Besides, studies have reported dual functionality of PLGA nanoparticles as both antigen delivery and adjuvant properties for enhancing immunity by promoting antigen engulfing by dendritic cells (DCs), activating and maturating DCs, and inducing effective immune response [27]”. See Page 2, line 74-78 which were marked in red.
- Sub-section of material and method section should be 2.1, 2.2, 2.3, ..... and so on.
Response to Question 5 of Reviewer #1
We have added the sub-section number in method part. See line 93, 116, 128, 141, 150, 185, 196, 222, 245, 256, 274, 283, which were marked in red in the revised manuscript.
- The title of the sub-section "Particle synthesis" should be "2.5. Preparation and characterization of PLGA nanoparticles"
Response to Question 6 of Reviewer #1
We have changed the sub-section title of 2.5 into “Preparation and characterization of FITC-EsxB-PLGA NPs”. Also, we re-arranged the orders of methodology part, in order to make the audiences easier to understand. See line 150, which were marked in red in the revised manuscript.
- It is suggested "Results" section should be divided into different sub-sections to make it more interesting and quick to understand for the reader.
Response to Question 7 of Reviewer #1
The "Results" section has been divided into different sub-sections. (See line 290,
308, 379, 422, which were marked in red in the revised version)
- Results of the "Particle synthesis" section (figure S3 and Table 1) should be included in the main manuscript. It should be deleted from supplementary information (figure S3 and Table 1) and included in the main manuscript (figure S3 and Table 1).
Response to Question 8 of Reviewer #1
Given that the main argument of this work is the establishment of antigen uptake quantification methodology, detailed description of the nanoparticle’s synthesis would cause distraction. In addition, the synthetic method is not new, it is based on the conventional amide reaction method [29]. We therefore left the nanoparticle synthesis and characterizations in the supporting information. See page 6-7, line 57-83, which were marked in red in the revised supporting information.
- It is important to include the value of the polydispersity index (PDI) while measuring the hydrodynamic diameter of (i) PLGA NPs and (ii) FITC-EsxB-PLGA NPs.
Response to Question 9 of Reviewer #1
We have added the PDI data of (i) PLGA NPs and (ii) FITC-EsxB-PLGA NPs in the supporting information in page 4 (Fig S4h).
- It is very important to include the "level of significance; p-value" in the results of supplementary table 1.
Response to Question 10 of Reviewer #1
The "level of significance; p-value" in the results of supplementary Table S1 has been added. See line 41-44, which were marked in red in the revised supporting information.
- The results of hydrodynamic diameter and zeta potential for FITC-EsxB-PLGA NPs in supplementary Figure S3 and supplementary Table 1 are different. Justify this difference.
Response to Question 11 of Reviewer #1
The data plotted in Figure S3 is a sample for presenting the distributions of diameter and zeta potential for FITC-EsxB-PLGA NPs. The data shown in Table S1 in the supporting information is the statistical results of multiple experiments. Thus, certain differences happened in Figure S3 and Table S1.
- It is important to elaborate on the discussion particularly related to the use of PLGA nanoparticles in vaccine delivery and related challenges and safety concerns. Also, incorporate the reason for low encapsulation efficiency (EF% - 30.7%) and loading capacity (LC% - 3.3%) in the current investigation and elaborate related discussion in the revised manuscript.
Response to Question 12 of Reviewer #1
PLGA is a biocompatible material that had been approved by the FDA for medical device application [2]. PLGA nanoparticles as the antigen carriers have excellent capacities for sustained release of antigen over a prolonged time, long systemic circulation half-lives, and the potential for targeting the APCs to achieve the higher maturation and antigen presentation of APCs [3].
The EE% is the ratio of the binding amount to the input amount of EsxB protein. In order to get more EsxB conjugated onto the PLGA nanoparticle surface, we increase the input amount of EsxB, leading to low EE%. Since the molecular weight of EsxB is about 11.5 kDa, the steric hindrance is large which lead to relative low LC%. Usually, the LC% of other nano delivery systems is about 1~10%, thus, in our study the LC% -
3.3% is reasonable.
The description above we have put in the supporting information where we showed the PLGA NPs characterization data. See page 6-7, line 57-83, which were marked in red in the revised supporting information.

Reviewer 2 Report
Comments and Suggestions for Authors
1. The abstract of the article must be supported with some key data obtained during the experiments.
2. It is not clear what was the route of administration of EsxB-PLGA NPs (50μg/mice) for immunization purpose? Was PBS used as solvent/dispersion medium for the NPs? What was the pH of PBS? Were the NPs sterilized prior to administration?
3. Why the endotoxin (LPS) was used to induce the of BMDC culture during the BMDC Cell characterization step?
4. There should be uniformity in mentioning the SI units, for example millilitre at some places was mentioned as “ml” whereas at most of the places the correct format “mL” was used.
5. To prepare the PLGA NPs, as mentioned “mixture buffer” (DCM: Acetone=3:2, v/v) was used, why DMSO was used in such a high volume ratio? What does “mixture buffer” mean?
6. 5% PVA of such a high molecular weight (Mw 31000 Da – 50000 Da) is too much to prepare the PLGA based NPs. Authors should optimize the concentration of PVA to get the desired characteristics in the developed NPs.
7. Correct the unit of time as “h” rather than “hrs” or “hours”.
8. This volume of DMSO used in the preparation cannot be evaporated in 4 h magnetic stirring that is too at room temperature condition.
9. During the chemical linkage of FITC-EsxB with PLGA NPs by condensation reaction between the carboxyl group on the NPs and the amino group of the EsxB. Firstly, the NPs (10 mg) was dissolved in 10 mL of 25 mmol/L MES buffer, PLGA NPs cannot be dissolved in MES buffer rather it can be dispersed/ suspended.
10. Similarly, if the activated NPs were again dissolved in 1 mL of 1 mg/mL EsxB solution. Then how the authors collected the FITC-EsxB-PLGA NPs for further studies?
11. Correct the equation for LE (%), there is no difference between the equations for EE and LE.
12. Line 143-144: The washing by centrifugation at 16500 g force at 4 ËšC, three times, might result the NPs of diameter ~180 nm, but how the shape was decided at this stage the FITC-EsxB-PLGA NPs were spherical? That is too without performing the electron microscopy (scanning / or transmission) to see the structural morphology of the NPs?
13. There must some characterization regarding the successful attachment of the FITC-EsxB with the NPs? How one can differentiate between the blank PLGA NPs and the FITC loaded or adsorbed NPs?
14. The conclusions must be supported with the data and some key findings of the result.
Author Response
Response letter
Dear Editor,
We now received the reviewers’ comments on our manuscript entitle “Uptake quantification of antigen carried by nanoparticles and its impact on carrier adjuvanticity evaluation” (vaccines-2662458). We are grateful to reviewers’ constructive suggestions, which help us to improve the manuscript. We have carefully revised the manuscript based on their comments and a list of changes made can be found in the following. All changes have been marked in the revised manuscript.
Response to the reviewers’ comments:
Reviewer #2
- The abstract of the article must be supported with some key data obtained during the experiments.
Response to Question 1 of Reviewer #2
The abstract has been revised according to the reviewer’s comments. (See line 24-
28, which were marked in red in the revised version)
- It is not clear what was the route of administration of EsxB-PLGA NPs (50 μg/mice) for immunization purpose? Was PBS used as solvent/dispersion medium for the NPs?
What was the pH of PBS? Were the NPs sterilized prior to administration?
Response to Question 2 of Reviewer #2
The subcutaneous injection was used to immunized the BALB/c mice with EsxB-PLGA NPs. PBS was used as dispersion medium for the NPs with the pH of 7.2. NPs have been sterilized prior to administration. The details were added in line 197 which were marked in red in the revised version.
- Why the endotoxin (LPS) was used to induce the of BMDC culture during the BMDC Cell characterization step?
Response to Question 3 of Reviewer #2
LPS is one of the commonly used stimuli to stimulate DCs maturation in vitro to initiate immune responses, which could be found in many literatures [e.g. Microbes Infect. 1999 Nov;1(13):1079-84.]. Immature DCs co-cultured with LPS could turn to mature DCs, thus up-regulate the DCs surface molecules such as MHCII and costimulatory molecules (CD80/86) while induce the secretion of chemokines, cytokines and proteases. The mature DC plays a role in the activate T cell polarization.
- There should be uniformity in mentioning the SI units, for example millilitre at some places was mentioned as “ml” whereas at most of the places the correct format “mL” was used.
Response to Question 4 of Reviewer #2
We have checked and corrected the SI units in the manuscript.
- To prepare the PLGA NPs, as mentioned “mixture buffer” (DCM: Acetone=3:2, v/v) was used, why DMSO was used in such a high volume ratio? What does “mixture buffer” mean?
Response to Question 5 of Reviewer #2
To prepare the PLGA NPs, mixed solvent of dichloromethane (DCM) and acetone was used with the ratio of DCM: Acetone=3:2, v/v. We did not use DMSO as the solvent.
Mixture buffer is the mixture of DCM and acetone.
- 5% PVA of such a high molecular weight (Mw 31000 Da – 50000 Da) is too much to prepare the PLGA based NPs. Authors should optimize the concentration of PVA to get the desired characteristics in the developed NPs.
Response to Question 6 of Reviewer #2
5% PVA with molecular weight of Mw 31000 Da – 50000 Da is commonly used in synthesis of PLGA nanoparticles, which could be found in many literatures [RSC Adv., 2020, 10, 4218-4231] [ Pharmaceutics. 2022 Apr; 14(4): 870]. In our experiment, we also used this method to synthesize the PLGA NPs. The results showed that the
PLGA NPs synthesized by this method showed good morphology and properties.
- Correct the unit of time as “h” rather than “hrs” or “hours”.
Response to Question 7 of Reviewer #2
We have corrected all in the revised manuscript. See line 156, 163, 205, 285,
331,which were marked in red in the revised version.
- This volume of DMSO used in the preparation cannot be evaporated in 4 h magnetic stirring that is too at room temperature condition.
Response to Question 8 of Reviewer #2
Actually, we did not use DMSO in the preparation of PLGA nanoparticle. DCM/Acetone mixture was used for dissolving the PLGA polymers, which has been clarified in the method part.
- During the chemical linkage of FITC-EsxB with PLGA NPs by condensation reaction between the carboxyl group on the NPs and the amino group of the EsxB. Firstly, the NPs (10 mg) was dissolved in 10 mL of 25 mmol/L MES buffer, PLGA NPs cannot be dissolved in MES buffer rather it can be dispersed/suspended.
Response to Question 9 of Reviewer #2
We have corrected the description of “dissolved” into “suspended”. (See page 4, line 166, which were marked in red in the revised version)
- Similarly, if the activated NPs were again dissolved in 1 mL of 1 mg/mL EsxB solution. Then how the authors collected the FITC-EsxB-PLGA NPs for further studies?
Response to Question 10 of Reviewer #2
The resulting FITC-EsxB-PLGA NPs were collected by centrifugation at 16500 g force at 4 ËšC.
- Correct the equation for LE (%), there is no difference between the equations for EE and LE.
Response to Question 11 of Reviewer #2
WFITC−EsxB on the NPs (mg)
EE(%) = WFITC−EsxB input(mg) × 100%
wFITC−EsxB on the NPs (mg)
LC (%) = WFITC−EsxB−PLGA NPs (mg) × 100%
These two equations did have differences. In order to make it more clearly, we put a detailed description below them. (See line 183-185, which were marked in red in the revised version)
- Line 143-144: The washing by centrifugation at 16500 g force at 4 ËšC, three times, might result the NPs of diameter ~180 nm, but how the shape was decided at this stage the FITC-EsxB-PLGA NPs were spherical? That is too without performing the electron microscopy (scanning / or transmission) to see the structural morphology of the NPs?
Response to Question 12 of Reviewer #2
SEM was employed to observe the morphologies of the NPs and the results was added as FigS4b and c in the revised supporting information.
- There must some characterization regarding the successful attachment of the FITCEsxB with the NPs? How one can differentiate between the blank PLGA NPs and the FITC loaded or adsorbed NPs?
Response to Question 13 of Reviewer #2
FTIR spectra was employed to demonstrate the successful synthesis of FITCEsxB-PLGA NPs, shown as FigS4i in the supporting information. In the FTIR spectra taken from samples of EsxB, PLGA NPs, and their corresponding EsxB conjugates, the bimodal due to the N-H stretching vibration in -NH2 in the pure EsxB (~3500 cm-1) disappeared in EsxB-PLGA NPs samples. Instead, the stretching vibration of -NH- (characteristic of the amide bond) can be found [4], suggesting the formation of amide bond, and thus successful binding of EsxB to the NPs.
The description was also added in the supporting information in line73-78, which were marked in red in the supporting information.
- The conclusions must be supported with the data and some key findings of the result.
Response to Question 13 of Reviewer #2
We have added some key data to the conclusion as follows, “According to the established method, we found that at the same antigen uptake amount, the antigen delivered by PLGA nanoparticles could elicited 3.6 times of IL-2 secretion (representative for cellular immune response activation) as well as 1.5 times of IL-12 secretion (representative for DC maturation level) compared with pure antigen feeding. Thus, by utilizing this method, we can easily discern whether the enhanced immune response is primarily due to increased antigen uptake or the adjuvanticity of the nanocarriers.” See page 13, line 499-505, which were marked in red in the conclusion.

Reviewer 3 Report
Comments and Suggestions for Authors
The manuscript aims to present an experimental method that allows to quantify antigen uptake carried by nanoparticles (PLGA) avoiding the artifacts due to chemical (nanoparticles) or biological (cells) environment when we use Fluorescence-based techniques. However, the most important aspect/ value of the paper is the scientific question behind the experimental work, that is to evaluate and to classify the role of nanoparticles as carrier and/or as an adjuvant (a very important point in vaccine formulation), and, in particular, if the adjuvant effect is only due to Ag or to nanoparticles as well. I really appreciated this intent of the authors. However, I have to point out two important issues on the paper and some minor observations listed below.
Major concerns:
1. The authors described the experimental approaches used to avoid artifacts due to Fluorescence-based technique as a necessary methodology for accurate uptake quantification. They have commented these approaches as a “methodology development”, as a kind of new approach. Why should these approaches be considered as new/good assay while they simply seem good laboratory practice, I mean, the right/correct way to performed the experiments. Please, explain well in the introduction or in the discussion the novelty the methodology used in the paper.
2. The authors based all the novelty and the importance of the results on the fact that the antigen uptake has been evaluated at the single-cell level. In my opinion, this is not the case. The investigation at single-cell level assumes techniques that the authors did not use (automated lab instrument for the study of some aspects – e.g., genomics, transcriptomics, proteomics, metabolomics and cell–cell interactions - at the single cell level. Despite the fact that the authors did not describe how they do the measurements at the single cell level (I suppose they divided the obtained fluorescence results for the number of the cells used in the assays, or not?), I suggest the authors to describe this aspect and to avoid to use a terminology as “at single cells level”, but to use “per number of cells” or something similar. Moreover, it should be considered that not all the cells are activated or phagocyte nanoparticles in vitro and not all in the same way, with the same extent. So, “single cell level” means that we are observing what’s happening exactly in a single cell.
Minor concerns:
1. Line 55. Please, specify the type of antigen and the type of nanoparticle used for the experiments, and the reason of this choice.
2. Line 75. Please, describe briefly the protocol used to isolate DC. Moreover, since in the reference cited (24) the authors used the cell sorting to separate the cells (so a different method respect to the manuscript), please, explain more in details how the macrophages were depleted and how the purity of DC was evaluated.
3. Line 120. How did the authors prove/test the successful fluorescence-labelling of protein antigen?
4. Line 146. The size distribution and zeta potential of nanoparticles are shown in supplementary figure S3. So, please, delete “can be measured” and write “were measured”. The same in line 148, I suppose that the EE and LC on the NPs were determined using a BCA protein quantification kit and not “can be measured”.
5. Line 153. Please, LC instead of LE.
6. Lines 151-154. Please, explain what means “W”, and please, check if the LC formula is correct at the denominator. Moreover, explain the meaning of the nominator and denominator of the two formulas for less expert readers.
7. Please, describe the results of all the figures (main and supplementary) throughout the manuscript where it is appropriate. So, add the description of those that are not mentioned in the current version of the manuscript.
8. Which was the concentration of FITC-EsxB used for the results shown in the Figure 2?
9. Please, clarify when the authors used DC from immunized or not immunized mice. Moreover, explain why the DC maturation has been performed using LPS for the Figure S1 and using antigen for the Figure 6. Please, comment coherently all these results.
Author Response
Response letter
Dear Editor,
We now received the reviewers’ comments on our manuscript entitle “Uptake quantification of antigen carried by nanoparticles and its impact on carrier adjuvanticity evaluation” (vaccines-2662458). We are grateful to reviewers’ constructive suggestions, which help us to improve the manuscript. We have carefully revised the manuscript based on their comments and a list of changes made can be found in the following. All changes have been marked in the revised manuscript.
Response to the reviewers’ comments:
Reviewer #3
The manuscript aims to present an experimental method that allows to quantify antigen uptake carried by nanoparticles (PLGA) avoiding the artifacts due to chemical (nanoparticles) or biological (cells) environment when we use Fluorescence-based techniques. However, the most important aspect/ value of the paper is the scientific question behind the experimental work, that is to evaluate and to classify the role of nanoparticles as carrier and/or as an adjuvant (a very important point in vaccine formulation), and, in particular, if the adjuvant effect is only due to Ag or to nanoparticles as well. I really appreciated this intent of the authors. However, I have to point out two important issues on the paper and some minor observations listed below.
Major concerns:
- The authors described the experimental approaches used to avoid artifacts due to Fluorescence-based technique as a necessary methodology for accurate uptake quantification. They have commented these approaches as a “methodology development”, as a kind of new approach. Why should these approaches be considered as new/good assay while they simply seem good laboratory practice, I mean, the right/correct way to performed the experiments. Please, explain well in the introduction or in the discussion the novelty the methodology used in the paper.
Response to Question 1 of Reviewer #3
Thank you very much for your helpful comments. We have added the explanation
in the introduction part, just in front of the introduction of our present research, shown as follows,
The correction of artefacts in common fluorescence-based characterizations is indeed a lab improvement. However, the protocol refers to the two step procedures of dividing the fed DC into two, one for trustworthy antigen uptake quantification and the other for immune response quantification, and then correlate the two to obtain the immune response per antigen uptake. See page 2, line 67-71 which were marked in red in the revised version.
- The authors based all the novelty and the importance of the results on the fact that the antigen uptake has been evaluated at the single-cell level. In my opinion, this is not the case. The investigation at single-cell level assumes techniques that the authors did not use (automated lab instrument for the study of some aspects – e.g., genomics, transcriptomics, proteomics, metabolomics and cell–cell interactions - at the single cell level. Despite the fact that the authors did not describe how they do the measurements at the single cell level (I suppose they divided the obtained fluorescence results for the number of the cells used in the assays, or not?), I suggest the authors to describe this aspect and to avoid to use a terminology as “at single cells level”, but to use “per number of cells” or something similar. Moreover, it should be considered that not all the cells are activated or phagocyte nanoparticles in vitro and not all in the same way, with the same extent. So, “single cell level” means that we are observing what’s happening exactly in a single cell.
Response to Question 2 of Reviewer #3
Indeed, the terminology as “at single cells level” is misleading. We have changed the description as “per number of cells” in the proper place of the revised manuscript as follows,
- See page 6, Line 252 in the manuscript. “The antigen feeding and uptake amount per numbers of cells was calculated by dividing the total antigen amount by the total number of cells.”
- See page 7, Line 287 in the manuscript. “The IL-12 secretion amount per numbers of cells was calculated by dividing the total IL-12 secretion by the total number of DCs.”
- See page 10, Line 390 in the manuscript. “Figure 4. Schematic for the quantification methodology of antigen uptake and IL-2 secretion per numbers of cells”
- See page 10, Line 397 in the manuscript. “We initially established a correlation between the amount of antigen uptake and the amount of antigen fed per numbers of
DCs.”
- See page 10, Line 400 in the manuscript. “By examining the triplicated results presented in Figure S8a-c, we observed a linear relationship between the amount of antigen fed and the amount of antigen uptake for both the free FITC-EsxB and FITC-
EsxB-PLGA NPs groups per numbers of cells.”
- See page 11, Line 419-421 in the manuscript. “Quantification of IL-2 secretion as a function of antigen uptake per numbers of cells in free antigen and antigen-NPs treatment groups.”
- See page 13, Line 495 in the manuscript. “The quantification of antigen uptake per numbers of cells relies on the release of fluorescent-labelled antigens from a specific environment to a standard one.”
- See page 13, Line 497-500 in the manuscript. “In this study, we have successfully developed a method to accurately quantify antigen uptake per numbers of cells. A linear relationship between antigen uptake and IL-2 secretion per numbers of cells is identified.”
- See page 13, Line 517-519 in the manuscript. “Figure S8: Quantification of FITC-EsxB or FITC-EsxB-PLGA NPs uptake per numbers of cells, Figure S9: Quantification of IL-2 secretion as a function of antigen feeding amount per numbers of cells, Fig.S10: Quantification of IL-2 secretion as a function of antigen uptake per numbers of cells.”
- See page 9, Line 110-111 in the supporting information. “Fig.S8.
Quantification of FITC-EsxB or FITC-EsxB-PLGA NPs uptake per numbers of cells.”
- See page 9, Line 116-117 in the supporting information. “Fig.S9.
Quantification of IL-2 secretion as a function of antigen feeding amount per numbers of cells.”
- See page 10, Line 122-123 in the supporting information. “Fig.S10.
Quantification of IL-2 secretion as a function of antigen uptake per numbers of cells.”
Minor concerns:
- Line 55. Please, specify the type of antigen and the type of nanoparticle used for the experiments, and the reason of this choice.
Response to Question 3 of Reviewer #3
The reason of the choice for antigen type and nanoparticle type have been added in the introduction. See page 2, line 72-83, which were marked in red in the revised version.
- Line 75. Please, describe briefly the protocol used to isolate DC. Moreover, since in the reference cited (24) the authors used the cell sorting to separate the cells (so a different method respect to the manuscript), please, explain more in details how the macrophages were depleted and how the purity of DC was evaluated.
Response to Question 4 of Reviewer #3
- The protocol to isolate DCs was shown as follow.
The mice were euthanized then steeped in the 70% ethanol solution for 5 min. The tibias and femurs were cut off and the muscles were removed under sterile conditions, then soaked in RPMI-1640 medium. Both ends of the bone were cut off with scissors, and the needle of a 1 mL syringe was inserted into the bone cavity to rinse the bone marrow cells out of the cavity into a sterile culture dish with RPMI-1640 medium. The cell suspension in the dish was collected and centrifuged at 300 g for 5 min, and the supernatant was discarded. The cell pellet was resuspended with Tris-NH4Cl red blood cell (RBC) lysis buffer to lyse the RBCs. Following the second centrifugation, the supernatant was discarded and the pelleted cells were washed with PBS and
collected.[30].
- How the macrophages were depleted and how the purity of DC was evaluated.
In the culture process, the cells, suspended in RPMI-1640 medium supplemented with 10% FBS, were distributed into 6-well non-treated culture surface plates at a density of 1×106 cell/mL. Subsequently, granulocyte colony-stimulating factor (GMCSF) was added into the medium to a final concentration of 20 ng/mL. The cells were cultured at 37 °C in an incubator containing 5% CO2. The culture medium was completely replaced 3 days later to remove the unattached cells and cell debris, then the fresh medium was supplemented with GM-CSF. On day 7, BMDC, the semisuspended cells and loosely attached cells, were collected by gently pipetting the medium against the plate, while BMDM were adherent on the plates (See line 98-115, which were marked in red in the revised version)
- Line 120. How did the authors prove/test the successful fluorescence-labelling of protein antigen?
Response to Question 5 of Reviewer #3
The successful fluorescence-labelling of protein antigen could be proved by the quantitation of protein: dye conjugation ratio (dye: protein or F/P molar ratio).
According to the protocol of fluorescence labelling [J Chem Biol. 2013 Jul; 6(3): 85– 95], EsxB (1mg/mL) was reacted with FITC as the weight ratio (1mg: 50μg) in 4 ℃ overnight, the final labeling reaction solution was dialyzed overnight to remove excess, nonconjugated dye. After that, the dialyzed conjugated was diluted 1/5 in
PBS, the absorbance of the conjugate at 280 nm (protein absorption peak) and 493 nm (FITC dye absorption peak) was measured. The molarity of the protein and the degree of labeling were calculated as the followed equation [https://assets.thermofisher.com/TFS-Assets/LSG/brochures/TR0031-Calc-FPratios.pdf]:
Protein concentration (M)= A280−(A493×CF) × dilution factor
ε
Moles dye per mole protein= A′max of the labeled protein × dilution factor ε ×protein concentration (M)
ε = protein molar extinction coefficient
CF = Correction factor; adjusts for the amount of absorbance at 280 nm caused by the FITC (0.1470) ε′ = molar extinction coefficient of the fluorescent dye (70,000 M-1 cm-1)
The results showed that the F/P ratio usually in the range of 2~4, which confirmed the successful fluorescence-labelling of protein antigen.
- Line 146. The size distribution and zeta potential of nanoparticles are shown in supplementary figure S3. So, please, delete “can be measured” and write “were measured”. The same in line 148, I suppose that the EE and LC on the NPs were determined using a BCA protein quantification kit and not “can be measured”.
Response to Question 6 of Reviewer #3
We have revised all the grammar errors and incorrected descriptions in the manuscript. (See line 171, 175, which were marked in red in the revised version.)
- Line 153. Please, LC instead of LE.
Response to Question 7 of Reviewer #3
We have corrected the typing errors in the manuscript. (See line 180, which were marked in red in the revised version.)
- Lines 151-154. Please, explain what means “W”, and please, check if the LC formula is correct at the denominator. Moreover, explain the meaning of the nominator and denominator of the two formulas for less expert readers.
Response to Question 8 of Reviewer #3
We have added detailed formula legends to explain the calculation. See in the revised methodology part in the revised manuscript. (See line 182-184, which were marked in red in the revised version.)
- Please, describe the results of all the figures (main and supplementary) throughout the manuscript where it is appropriate. So, add the description of those that are not mentioned in the current version of the manuscript.
Response to Question 9 of Reviewer #3
We have checked all the figures throughout the manuscript and added description in the revised manuscript and supporting information. In order to make the manuscript logically, we added the description of BMDC characterization, the expression and concentration of EsxB, nanoparticle characterization, and the serum EsxB-specific IgG titre with EsxB-PLGA NPs in the supporting information. See line 15-23, 28-29, 37-38, 58-84, 98-101, which were marked in red in the supporting information.
- Which was the concentration of FITC-EsxB used for the results shown in the Figure 2?
Response to Question 10 of Reviewer #3
In Figure 2a, the fluorescence intensity of FITC-EsxB at a concentration of 5 µg/mL was determined in different solvents (PBS buffer, 10% SDS buffer, and PBS/DMSO buffer (1﹕2, v/v). In Figure 2b, the fluorescence of free FITC-EsxB at a feeding concentration of 1 mg/mL was determined in/released from endosomes. In Figure 2c, the fluorescence of FITC-EsxB-PLGA NPs at a concentration of 200 µg/mL was determined in/release from NPs surface.
The details of concentration in Figure 2 have been added in the revised manuscript.
(See line 328-334, which were marked in red in the revised version)
- Please, clarify when the authors used DC from immunized or not immunized mice.
Moreover, explain why the DC maturation has been performed using LPS for the Figure S1 and using antigen for the Figure 6. Please, comment coherently all these results.
Response to Question 11 of Reviewer #3
The DCs used in the experiments were all from non-immunized mice, the spleen cells were extracted from immunized mice.
The DC maturation could be induced by many factors, including LPS stimulation and antigen uptake (Fig.S1a). In the Figure S1d, we used LPS stimulus to induce DC maturation in vitro. Immature DCs co-cultured with LPS could turn to mature DC thus up-regulate the DCs surface molecules such as MHCII and co-stimulatory molecules (CD80/86). In the Figure 6, we investigated DC maturation after stimulation by the
PLGA NPs and EsxB-PLGA NPs and EsxB as positive control.

Reviewer 4 Report
Comments and Suggestions for Authors
This manuscript proposed the quantification methods of cellular uptake of antigen and correlation with adjuvant property of antigen carriers. Th authors checked the fluorescence f FITC-labeled antigen in various medium and fluorescence when antigen was extracted from cells. Normalized fluorescence intensity was calculated from these data for estimation of adjuvant property of PLGA nanoparticles themselves. The approach of this manuscript is unique and will provide useful insight to know the adjuvant property of antigen carriers. Some minor corrections would further increase the scientific quality of this manuscript. The list of comments is as follows:
-The authors should explain what EsxB is and how the authors obtained EsxB.
-Figure 5b and 6b: The authors should explain more clearly how “per DC” was measured for calculation of data within these Figures.
-Lines 382-383: Is “no significant difference” correct compared with Figure 1 and 2a?
-Figures S6-S9: The authors should explain why intercept value of each fitted curve did not become zero.
-The authors should discuss whether technique proposed within this manuscript can be applicable into other antigen carriers such liposome, micelle, exosome and so on.
Author Response
Response letter
Dear Editor,
We now received the reviewers’ comments on our manuscript entitle “Uptake quantification of antigen carried by nanoparticles and its impact on carrier adjuvanticity evaluation” (vaccines-2662458). We are grateful to reviewers’ constructive suggestions, which help us to improve the manuscript. We have carefully revised the manuscript based on their comments and a list of changes made can be found in the following. All changes have been marked in the revised manuscript.
Response to the reviewers’ comments:
Reviewer #4
This manuscript proposed the quantification methods of cellular uptake of antigen and correlation with adjuvant property of antigen carriers. The authors checked the fluorescence FITC-labeled antigen in various medium and fluorescence when antigen was extracted from cells. Normalized fluorescence intensity was calculated from these data for estimation of adjuvant property of PLGA nanoparticles themselves. The approach of this manuscript is unique and will provide useful insight to know the adjuvant property of antigen carriers. Some minor corrections would further increase the scientific quality of this manuscript. The list of comments is as follows:
- The authors should explain what EsxB is and how the authors obtained EsxB.
Response to Question 1 of Reviewer #4
EsxB is a protein secreted by S. aureus. The EsxB was prepared using standard
IPTG induction protocol, see attached in the section of “expression and purification of EsxB antigen” in the methodology part. (See page 3-4, line 129-140, which were marked in red in the revised version). The obtained EsxB protein was also determined by SDS-PAGE, shown as FigS2.
- Figure 5b and 6b: The authors should explain more clearly how “per DC” was measured for calculation of data within these Figures.
Response to Question 2 of Reviewer #4
The calculation method of “per DC” have been added in the “Materials and Mothods” parts (2.9 and 2.10) in the revised manuscript. See page 6, line 252-253, 262264, 268-273 in the method part.
- Lines 382-383: Is “no significant difference” correct compared with Figure 1 and 2a? Response to Question 3 of Reviewer #4
The Figure 1 plotted the fluorescence intensity of FITC in different solvents at the concentration of 5 μg/mL. Figure 2a plotted the fluorescence intensity of FITC labeled
EsxB (FITC-EsxB) in different environments at the EsxB concentration of 5 μg/mL.
- Figures S6-S9: The authors should explain why intercept value of each fitted curve did not become zero.
Response to Question 4 of Reviewer #4
In the prime manuscript, the Figure S6a plotted the standard curve of FITC-EsxB by BCA method. In the BCA regent kits, the CuSO4 solution, presenting green color, could cause certain absorption in UV-spectra without any protein adding. This absorbance caused by background solution leads to the result of intercept value, which is impossible to become zero.
The Figure S7 in the supporting information have been re-fitted, shown as Figure
S8 in the revised version.
In Figure S8 and S9, since spleen cells were used in the experiment, which secrets IL-2 in the normal status. Thus, even without antigen feeding, the IL-2 secretion is objective existed, leading to certain values in IL-2 determination.
- The authors should discuss whether technique proposed within this manuscript can be applicable into other antigen carriers such liposome, micelle, exosome and so on.
Response to Question 5 of Reviewer #4
Since the PLGA NPs is a model system we chose to deliver the model antigen EsxB, not a specific choice, the established methodology could also be applied in other antigen carriers, such as liposome, micelle, exosome and so on. We have added this statement into the main text. See line 490-492, which were marked in red in the revised version.

Reviewer 5 Report
Comments and Suggestions for Authors
In the manuscript titled “uptake quantification of antigen carried by nanoparticles and its impact on carrier adjuvanticity evaluation” Zhu and coauthors developed approach to solve one of the important problems in nanoparticles delivery, namely inconsistency of fluorescence-based signal readout under different chemical and biological environments. The authors quantified a series of fluorescence dye labeled antigen under different environmental conditions and figured out a consistent method for antigen quantification. This was then used to determine the levels of immune activation by either intrinsic immune activation by the nanoparticle carrier and enhanced antigen uptake.
This paper is practical research which address one of the important questions regarding antigen delivery by nanoparticles. The experiments were carefully designed and the results are convincing. The reviewer believed that the current manuscript meets the publication requirements of Vaccines. Some minor concerns:
1. Although a method to comprehensively evaluate adjuvanticity is lacking, only looking at certain surface molecule expressions and certain cytokines might not be enough to claim adjuvanticity evaluation. For example, the cellular arm of the immune response is not addressed in the manuscript.
2. What about other nanoparticle carrier systems? At least one other than PLGA will be helpful for the paper’s claims.
Author Response
Response letter
Dear Editor,
We now received the reviewers’ comments on our manuscript entitle “Uptake quantification of antigen carried by nanoparticles and its impact on carrier adjuvanticity evaluation” (vaccines-2662458). We are grateful to reviewers’ constructive suggestions, which help us to improve the manuscript. We have carefully revised the manuscript based on their comments and a list of changes made can be found in the following. All changes have been marked in the revised manuscript.
Reviewer #5
Response to the reviewers’ comments:
In the manuscript titled “uptake quantification of antigen carried by nanoparticles and its impact on carrier adjuvanticity evaluation” Zhu and coauthors developed approach to solve one of the important problems in nanoparticles delivery, namely inconsistency of fluorescence-based signal readout under different chemical and biological environments. The authors quantified a series of fluorescence dye labeled antigen under different environmental conditions and figured out a consistent method for antigen quantification. This was then used to determine the levels of immune activation by either intrinsic immune activation by the nanoparticle carrier and enhanced antigen uptake.
This paper is practical research which address one of the important questions regarding antigen delivery by nanoparticles. The experiments were carefully designed and the results are convincing. The reviewer believed that the current manuscript meets the publication requirements of Vaccines. Some minor concerns:
- Although a method to comprehensively evaluate adjuvanticity is lacking, only looking at certain surface molecule expressions and certain cytokines might not be enough to claim adjuvanticity evaluation. For example, the cellular arm of the immune response is not addressed in the manuscript.
Response to Question 1 of Reviewer #5
The reviewer's opinion is accurate. We should indeed study more measures of immune response activation to determine the universality of our approach. In our follow-up studies, we will have more quantitative studies on the relationship between immune response and antigen uptake.
- What about other nanoparticle carrier systems? At least one other than PLGA will be helpful for the paper’s claims.
Response to Question 2 of Reviewer #5
We have studied the SiO2 nanoparticles as antigen delivery systems by our established method. We did not put the data in this paper, but will publish it in another work, which studies the impact of SiO2 nanoparticles morphologies on immune response activation.

Round 2
Reviewer 1 Report
Comments and Suggestions for Authors
The revised manuscript improved well.
Comments on the Quality of English LanguageMinor editing of the English language is required
Author Response
Please see the attachment.
Reviewer #1
The revised manuscript improved well.
- Minor editing of the English language is required.
Response to Question 1 of Reviewer #1
We have revised all the grammar errors and incorrected descriptions in the manuscript.

Reviewer 2 Report
Comments and Suggestions for Authors
Authors have revised the manuscript. The manuscript may be accepted after responding the following queries:
1. Check out the unit of centrifugation. Centrifugation at 16500 g force, is it correct?
2. In Figure 3, where ***P<0.001 and ****P < 0.0001 were expressed?
3. In Figure 5, where is the need of ****p < 0.001?
4. Figure 6, needs the expression of *P<0.05, **P <0.01, ***P<0.001, ****P < 0.0001 expressions are needed in the caption of Figure 6, but these are missing here, while there was no need to mention “****p < 0.001” in Figure 5, but it is there.
5. First round comment #10: “Similarly, if the activated NPs were again dissolved in 1 mL of 1 mg/mL EsxB solution. Then how the authors collected the FITC-EsxB-PLGA NPs for further studies?”
Authors response to Question 10: The resulting FITC-EsxB-PLGA NPs were collected by centrifugation at 16500 g force at 4 ËšC.
My question was, if the NPs were dissolved EsxB solution, then where was the integrity of NPs remained? Then how the authors mentioned that the “NPs” were collected centrifugation?
6. Why we call the mixture of DCM and acetone as mixture “buffer”, why not its solvent mixture?
7. What was the ratio of PBS/DMSO during the influence of different chemical environments on FITC fluorescence intensity?
Author Response
Please see the attachment.
Authors have revised the manuscript. The manuscript may be accepted after responding the following queries:
- Check out the unit of centrifugation. Centrifugation at 16500 g force, is it correct?
Response to Question 1 of Reviewer #2
We have changed the unit of centrifugation as “´g”. See page 3 line 101. Page 4, line 153 161, 163. Page 5, line 229. Page 6, line 262. These changes have been marked in red.
- In Figure 3, where ***P<0.001 and ****P < 0.0001 were expressed?
Response to Question 2 of Reviewer #2
Statistical analysis differences ***P<0.001 and ****P<0.0001 were unmentioned in Figure 3, thus we removed them for avoiding further confusions.
- In Figure 5, where is the need of ****p < 0.001?
Response to Question 3 of Reviewer #2
Statistical analysis difference ****p < 0.001 was unmentioned in Figure 5, thus have been removed.
- Figure 6, needs the expression of *P<0.05, **P <0.01, ***P<0.001, ****P < 0.0001 expressions are needed in the caption of Figure 6, but these are missing here, while there was no need to mention “****p < 0.001” in Figure 5, but it is there.
Response to Question 4 of Reviewer #2
We have added the statistical differences in the caption of Figure 6 and removed the unmentioned statistical differences in Figure 5.
- First round comment #10: “Similarly, if the activated NPs were again dissolved in 1 mL of 1 mg/mL EsxB solution. Then how the authors collected the FITC-EsxB-PLGA NPs for further studies?”
Authors response to Question 10: The resulting FITC-EsxB-PLGA NPs were collected by centrifugation at 16500 g force at 4 ËšC.
My question was, if the NPs were dissolved EsxB solution, then where was the integrity of NPs remained? Then how the authors mentioned that the “NPs” were collected centrifugation?
Response to Question 5 of Reviewer #2
We have revised the incorrect description of the NPs existence in EsxB solution. The PLGA NPs was “suspended” in EsxB solution, rather than “dissolved”. Thus, the PLGA NPs did not dissociate in EsxB solution. After conjugation, the FITC-EsxB-PLGA NPs were collected by centrifugation at 16500 ´g at 4 ËšC. See page 4, line 162.
- Why we call the mixture of DCM and acetone as mixture “buffer”, why not its solvent mixture?
Response to Question 6 of Reviewer #2
We have changed the description of “mixture buffer” for “mixed solvents” to avoid misunderstanding. See page 4, line 148.
- What was the ratio of PBS/DMSO during the influence of different chemical environments on FITC fluorescence intensity?
Response to Question 7 of Reviewer #2
The ratio of PBS/DMSO buffer was 1:2(v/v) during the influence of different chemical environments on FITC fluorescence intensity. See 6, line 290.
